# Utilization of Fly Ash and Red Mud in Soil-Based Controlled Low Strength Materials

Xianghui Kong [1,*], Gaoqiang Wang [1], Shu Rong [2], Yunpeng Liang [1], Mengmeng Liu [1] and Yanhao Zhang [3,*]

1   School of Transportation Engineering, Shandong Jianzhu University, Jinan 250101, China;
    wanggq97@163.com (G.W.); liangyunpeng99@163.com (Y.L.); liumengmeng992022@163.com (M.L.)
2   Jinan Kingyue Highway Engineering Company Limited, Jinan 250014, China; rongs2019@163.com
3   School of Municipal & Environmental Engineering, Shandong Jianzhu University, Jinan 250101, China
*   Correspondence: kongxh@sdjzu.edu.cn (X.K.); zhyh@sdjzu.edu.cn (Y.Z.)

**Abstract:** To reduce the harm caused by industrial solid waste to the environment, and solve the problem of excavated soil disposal in buried pipeline management projects, this study proposes a method to produce soil-based controlled low strength materials (CLSM) by using industrial solid wastes (including fly ash and red mud) as partial replacements for cement. The properties of CLSM were characterized in terms of flowability, unconfined compressive strength, phase composition and microstructure. The test results showed that fly ash could significantly improve the flowability of CLSM, while red mud had more advantages for the strength development. When 20% fly ash and 30% red mud were combined to replace cement, the fluidity of CLSM was 248 mm, and the unconfined compressive strength (UCS) at 3, 7 and 28 days was 1.08, 1.49 and 3.77 MPa, respectively. The hydration products of CLSM were mainly calcium silicate hydrate gels, ettringite and calcite. Fly ash provided nucleation sites for cement hydration, while the alkali excitation of red mud promoted the dissolution of $SiO_2$ and $Al_2O_3$ in fly ash. The filling and gelation of hydration products make the microstructure dense, which improves the mechanical properties of the mixture.

**Keywords:** controlled low strength materials; fly ash; red mud; excavated soil; mix design; microscopic properties





## 1. Introduction

With the development of municipal pipeline management projects, a large amount of spoil generated from construction sites in various regions cannot be effectively resolved. Excavated soil for reuse is considered to have a positive impact on both the engineering economy and the environment [1–3]. The construction of traditional backfill materials has problems such as small operation space and a dead angle on the contact surface with the pipeline, which makes it difficult to guarantee the compaction quality [4]. Therefore, it is important to develop a backfill material with compactness and ease of construction by using excavated soil.

Controlled low strength materials (CLSM), also known as flow backfill materials with self-leveling and self-compacting properties, can replace traditional compacted backfill [5,6]. CLSM is often applied to bridge approaches, retaining wall backs, pipe trench backfilling and other projects [7]. In general, the flowability of CLSM needs to be between 200 and 300 mm, and the 28-day UCS is required to be less than 8.3 MPa [8]. The strength can be controlled effectively by adjusting the dosage of cement. Considering excavability, the UCS strength is no higher than 2.1 MPa [9]. CLSM does not require high strength, so a large number of non-standard materials are allowed in its production [10].

Currently, the mix proportion of CLSM is not constrained [11]. Wu and Tsai [12] used rubber particles as fine aggregate instead of traditional sand and found that non-sand CLSM was not flowable. Kuo and Gao [13] developed a CLSM using the bottom ash of a municipal solid waste incinerator as aggregate which had sufficient load-bearing capacity with 28-day UCS of 6.25 ~ 5.87 MPa. Yan et al. [14] proposed to produce CLSM with

dredged sediment and bottom ash. Their study showed that the formation of calcium silicate hydrate (C-S-H) gels was the main reason for the strength improvement, which also had a positive effect on heavy metals immobilization. The CLSM they produced had a good general performance while achieving a waste utilization rate of 80%. Kim et al. [15] used steel slag as a new aggregate to develop CLSM. They found that the flowability and strength of CLSM decreased with the increase of steel slag content. They also pointed out that steel slag particle size also has a different effect on the properties of the mixture.

Cement hydration produces a large amount of cementitious material that allows CLSM to set and harden [16]. However, the production of cement is a process with high energy consumption and high emissions. Therefore, the application of industrial solid waste as supplementary cementitious material (SCM) in CLSM is also attracting attention. The inclusion of industrial solid waste in CLSM production is intended to control performance development while reducing production costs [17]. Fly ash has been proved to be effective in improving the fluidity of CLSM, which is greatly related to its spherical shape [18]. Moreover, fly ash can improve the strength of mixture by reducing porosity and improving the interlocking mechanism between aggregates, especially in the late curing period [19,20]. Alkaline activation is helpful to stimulate the vitality of fly ash and improve the strength of mixture [21]. However, other admixtures are sometimes added to CLSM to meet engineering applications, such as those with a rapid hardening property.

Red mud is a solid waste produced by the aluminum industry, with an annual output of up to 70 million tons in China [22]. For a long time, the strong alkalinity and high content of heavy metals in red mud have made it challenging to utilize. However, the main components of red mud contain many volcanic ash active substances such as $SiO_2$ and $Al_2O_3$, and the high alkalinity of red mud can also be used as a source of alkali to activate cementitious materials [23]. Cheng et al. [24] developed a new type of cementitious material using red mud and evaluated its physical and chemical properties. The new cementitious material took a shorter time to reach the peak heat of hydration and its microstructure was more compact. After adding red mud, the voids of the cementitious material were filled and the mechanical properties were improved. This bonding property of red mud means it was thought it could be added to the production of CLSM as SCM [25,26].

In this study, fly ash and red mud were used as replacement materials for cement, and excavated soil was used as fine aggregate to produce the CLSM mixture. Twenty different mix ratios were used to explore the effects of fly ash and red mud replacing cement on the macro properties of CLSM. The alternative types were divided into independent alternative and co-alternative. Taking high fluidity and early strength required by CLSM as control indexes, the optimal ratio of fly ash and red mud to replace cement was studied. More importantly, the interaction mechanism between fly ash and red mud is analyzed by microcosmic means. This provides a new idea for the utilization of fly ash and red mud resource engineering.

## 2. Materials and Methods

### 2.1. Raw Materials

In this study, excavated soil was used as fine aggregate, cement was used as the main cementing agent, and fly ash and red mud were used as supplementary cementitious materials. Excavated soil was taken from a municipal engineering pipeline construction in Shandong Province. After removing vegetation and debris, the soil was sealed in bags and placed in a cool room to reduce the effect of ambient temperature and humidity on the soil. Excavated soil was tested for particle analysis and basic physical properties according to JTG 3430-2020 [27]. Figure 1 shows the grain size distribution of natural soil. Other physical properties of the soil are summarized in Table 1.

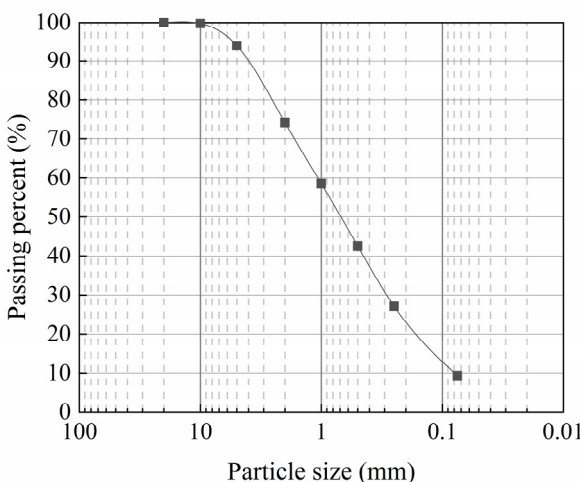

**Figure 1.** Grain size distribution of the natural soil.

**Table 1.** Physical properties of soil.

| Physical Properties | Value |
| --- | --- |
| Natural water content (%) | 10.8 |
| Liquid limit (%) | 48.9 |
| Plastic limit (%) | 22.9 |
| Plasticity index | 26.0 |
| Specific gravity | 2.74 |
| Particles (< 0.075 mm) (%) | 9.34 |

The cement was P.O 42.5 cement produced by a company in Shandong Province (Jiuqi Building Materials Co., LTD, Shandong, China), and its specific surface area was 0.36 m$^2$/g. As the main cementing material, the hydration of cement produces a large number of cementing substances which could improve the strength of the mixture. The fly ash used in this study was taken from a power plant in Henan Province. The specific surface area of the fly ash was 0.43 m$^2$/g. The red mud was produced by the industrial Bayer aluminum process from a red mud dump in Zhengzhou City, Henan Province (Yixiang New Materials Co., LTD, Zhengzhou, China). Its specific surface area was 0.68 m$^2$/g and the pH value was 12.57. The appearances of raw materials and the microscopic images of the particle surface are shown in Figure 2.

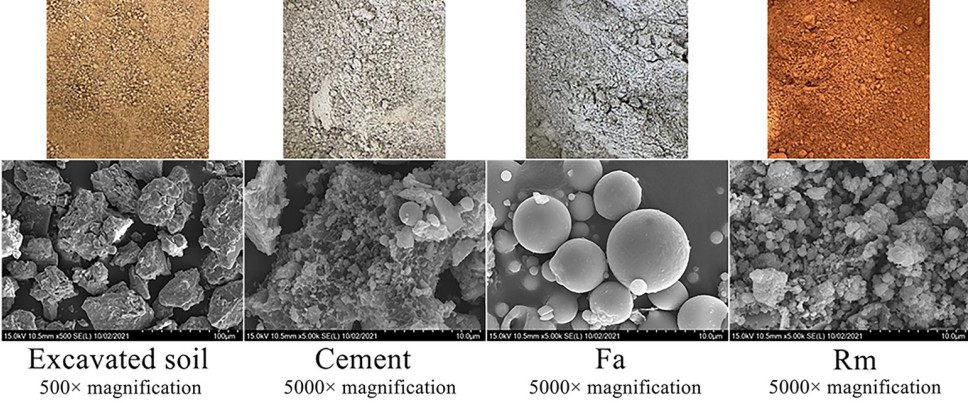

**Figure 2.** Appearance and microscopic images of raw materials.

The chemical compositions of the raw materials are listed in Table 2, which were measured by X-ray fluorescence (XRF, S8 TIGER, Bruker Corporation, Karlsruhe, Germany). According to the measurement results, it can be seen that the compositions of excavated

soil were mainly $SiO_2$ and $Al_2O_3$, but the cement was dominated by CaO and $SiO_2$. The high content of $SiO_2$ and $Al_2O_3$ in fly ash and red mud gave them the ability to partially replace cement.

**Table 2.** Chemical compositions of excavated soil, cement, fly ash and red mud.

| Compound | Excavated Soil (%) | Cement (%) | Fa (%) | Rm (%) |
|---|---|---|---|---|
| $SiO_2$ | 70.41 | 31.40 | 54.90 | 28.66 |
| CaO | 1.50 | 42.10 | 3.96 | 3.37 |
| $Al_2O_3$ | 15.92 | 12.70 | 28.10 | 23.32 |
| $Fe_2O_3$ | 4.59 | 4.19 | 5.38 | 26.19 |
| $K_2O$ | 2.77 | 1.17 | 2.94 | 1.42 |
| $Na_2O$ | 1.58 | 0.62 | 0.78 | 12.40 |
| MgO | 1.95 | 3.28 | 0.77 | 0.32 |
| $TiO_2$ | 0.78 | 0.66 | 1.44 | 1.91 |
| MnO | 0.08 | 0.18 | 0.06 | 0.06 |
| ZnO | 0.01 | 0.01 | / | 0.01 |
| $SO_3$ | 0.07 | 3.32 | 0.68 | 1.84 |

### 2.2. Sample Preparation and Mix Design

All raw materials needed to be dried and weighed by a digital balance with 0.1 g accuracy. The excavated soil used was passed through a 5 mm mesh screen. The soil, cement, fly ash and red mud were put into the mixer and dry mixed for 30 s. Subsequently, calculated water was added to them and mixing was continued for 2 min. After mixing was completed, the CLSM slurry was partially poured into the flowability test tool and the remaining slurry was molded in a 70.7 mm × 70.7 mm × 70.7 mm cube test mold. The specimen prepared for molding was pre-cured at room temperature for 24 h and then demolded. The samples were conditioned in a curing box at a temperature of 20 °C and 97% humidity.

A comprehensive test program was designed to evaluate the effect of fly ash and red mud on CLSM, as shown in Table 3. A total of 20 different mix designs are included. The presence of Fa0 (or Rm0) in the named species of the mixture means that the CLSM performance characteristics when red mud (or fly ash) is separately studied. The amount of water and soil in all mixtures was fixed. The replacement amount of fly ash was increased from 0% to 30%, and the replacement amount of red mud was increased from 0% to 40%. The replacement amount is the proportion of the mass of fly ash or red mud replacing the mass of cement.

### 2.3. Testing Methods
#### 2.3.1. Flowability Test

The flowability of CLSM was tested according to ASTM D 6103 [28]. A cylinder with a height of 150 mm and an inner diameter of 75 mm was used. The cylinder was lifted as soon as the CLSM slurry filled it and the mixture was allowed to diffuse freely on the plane. When the diffusion range was no longer increasing, the watershed size was measured with a ruler in both vertical directions. The average of the two measurements was specified as the fluidity of the CLSM. The accuracy was 1 mm.

#### 2.3.2. Unconfined Compressive Strength Test

The UCS test for CLSM was performed in accordance with JGJ/T 70-2009 [29]. The curing ages of test samples were 3, 7 and 28 days, respectively. The test instrument was a pavement material strength tester (LD127, Guanghui test instrument co., LTD, Cangzhou, China), with the accuracy of 0.01 MPa. The base plate was pressurized at a lifting rate of 1 mm per minute on the sample, and the maximum strength value of the sample at the time

of destruction was recorded. Each test included three specimens, and the average value of the test results of the three specimens was taken as the final value.

**Table 3.** Mix proportions for CLSM mixtures.

| Mix Name | Fa/Binder (%) | Rm/Binder (%) | Weight Fraction | | | | |
| --- | --- | --- | --- | --- | --- | --- | --- |
| | | | Water | Soil | Binder | | |
| | | | | | Cement | Fa | Rm |
| Fa0-Rm0 | | 0 | | | 0.29 | 0 | 0 |
| Fa0-Rm10 | | 10 | | | 0.26 | 0 | 0.03 |
| Fa0-Rm20 | 0 | 20 | 0.58 | 1 | 0.23 | 0 | 0.06 |
| Fa0-Rm30 | | 30 | | | 0.20 | 0 | 0.09 |
| Fa0-Rm40 | | 40 | | | 0.17 | 0 | 0.12 |
| Fa10-Rm0 | | 0 | | | 0.26 | 0.03 | 0 |
| Fa10-Rm10 | | 10 | | | 0.23 | 0.03 | 0.03 |
| Fa10-Rm20 | 10 | 20 | 0.58 | 1 | 0.20 | 0.03 | 0.06 |
| Fa10-Rm30 | | 30 | | | 0.17 | 0.03 | 0.09 |
| Fa10-Rm40 | | 40 | | | 0.15 | 0.03 | 0.12 |
| Fa20-Rm0 | | 0 | | | 0.23 | 0.06 | 0 |
| Fa20-Rm10 | | 10 | | | 0.20 | 0.06 | 0.03 |
| Fa20-Rm20 | 20 | 20 | 0.58 | 1 | 0.17 | 0.06 | 0.06 |
| Fa20-Rm30 | | 30 | | | 0.15 | 0.06 | 0.09 |
| Fa20-Rm40 | | 40 | | | 0.12 | 0.06 | 0.12 |
| Fa30-Rm0 | | 0 | | | 0.20 | 0.09 | 0 |
| Fa30-Rm10 | | 10 | | | 0.17 | 0.09 | 0.03 |
| Fa30-Rm20 | 30 | 20 | 0.58 | 1 | 0.15 | 0.09 | 0.06 |
| Fa30-Rm30 | | 30 | | | 0.12 | 0.09 | 0.09 |
| Fa30-Rm40 | | 40 | | | 0.09 | 0.09 | 0.12 |

### 2.3.3. XRD Test

X-ray diffraction (XRD, PANalytical B.V., EA Almelo, The Netherlands) was used to identify the phase composition. The samples for XRD were taken from samples cured for 28 days. Samples were dried and passed through a 0.075 mm mesh screen. A Cu target was used for $5°$ to $90°$ scanning at $10°$ per minute.

### 2.3.4. SEM Test

The microstructure of the mixture determines the macroscopic properties of the sample. Therefore, a scanning electron microscope (SEM, SU5000, Hitachi High-tech Company, Tokyo, Japan) was used to analyze the CLSM microstructure. The samples were cured for 28 days and cut into 10 mm × 5 mm × 5 mm block shapes. The spray-gold process was necessary because of the weak conductivity of the samples.

## 3. Results and Discussion

### 3.1. Flowability Analysis

Self-leveling properties require the high flowability of CLSM mixtures. The effects of fly ash and red mud on the flowability of CLSM are shown in Figure 3. From Figure 3, it can be seen that the increase of fly ash content improved the flowability of the mixture, while the red mud had the opposite effect. The flowability of Fa20-Rm0 was the largest among all the mix proportion samples. In this replacement, the ball of fly ash played an effective role in promoting the flowability increase. However, the excessive admixture was not conducive to the development of CLSM workability. Fly ash replacement continued to increase, resulting in a lower flowability. As the amount of red mud replacement increased from 0% to 40%, the mixture flowability decreased from 230 to 175 mm. Fluidity below the minimum limit (200 mm) occurred at 30% and 40% red mud replacement. The explanation for this phenomenon is that the specific surface area of red mud was much larger than

that of cement, and the increase in the amount of water adsorbed on the surface led to a decrease in the flowability of the mixture. The free water in CLSM was reduced by the replacement of cement with red mud, hence more water was required in the mix to meet the high flowability requirement.

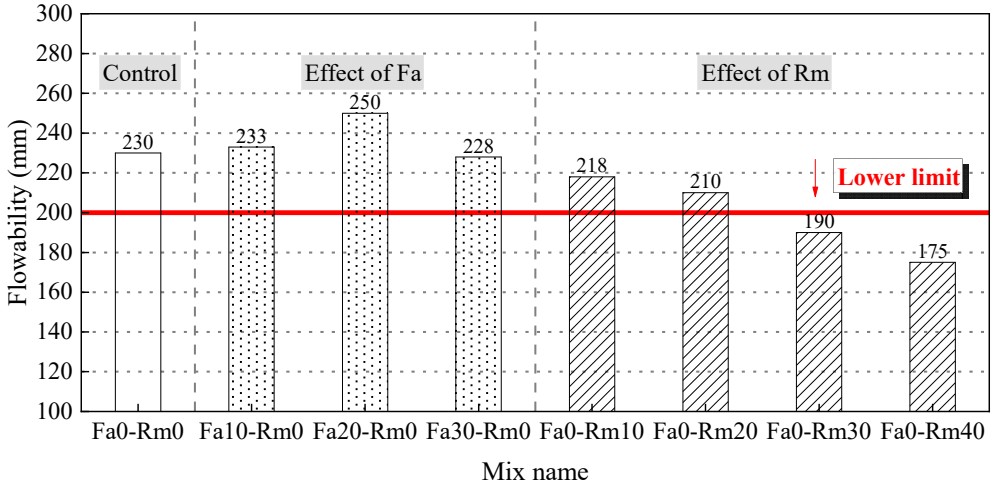

**Figure 3.** Flowability of CLSM with independent replacement of Fa and Rm.

The effects of combined fly ash and red mud replacement cement on CLSM flowability are shown in Figure 4. According to the trend of the curves in the figure, it can be seen that the effects of fly ash and red mud on flowability under combined replacement had the same pattern as that in independent replacement. The data comparison revealed that the difference in fluidity between Fa10-Rm10 and Fa10-Rm40 was 38 mm, between Fa20-Rm10 and Fa20-Rm40 was 25 mm, and between Fa30-Rm10 and Fa30-Rm40 was 35 mm. Therefore, in spite of the presence of red mud, the 20% fly ash replacement amount still had a significant improvement in flowability. Only Fa10-Rm40, Fa30-Rm30 and Fa30-Rm40 had flowability below the limit. This was due to the fact that fly ash replacement at 10% and 30% had less effect on the increase of fluidity than replacement at 20%. Combined with the high red mud replacement, the superimposed effect of the double negative effect reduced the flowability. It should be noted that the CLSM produced in this research used 100% soil as fine aggregate. Compared with sand, soil-based CLSM is more prone to segregation at high fluidity and is unable to harden the mixture quickly.

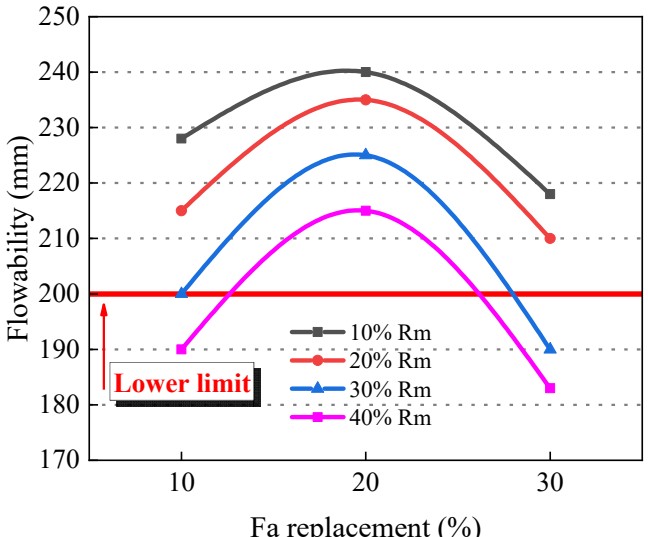

**Figure 4.** Flowability of CLSM with co-replacement of Fa and Rm.

### 3.2. UCS Analysis

The effects of independent replacement of fly ash and red mud on the strength of CLSM are shown in Figure 5. It was a common phenomenon that the strength of all mixtures increased with curing age. Analysis of the test results revealed that the CLSM strength kept decreasing as the amount of fly ash replacement increased. For instance, the 3, 7 and 28 days UCS of Fa20-Rm0 were 0.62, 1.09 and 3.73 MPa. Compared with the control group (Fa0-Rm0), the strength decreased by 45%, 39% and 19%, respectively. CLSM strength increased first and then decreased with the increase in red mud replacement amount. The UCS of Fa0-Rm30 always remained maximum at all ages. The 3, 7 and 28 days UCS of Fa0-Rm30 were 1.25, 2.24 and 5.11 MPa, respectively. The strength growth rates were 11%, 26% and 12% compared with Fa0-Rm0. Based on the strength change, there was a threshold value for the red mud content in CLSM, and 30% was the optimal amount of replacement when red mud was independently affected. The most significant factor affecting the strength change is the content of cement in the mixture, thus controlling the cement content is an important way to achieve controllability of CLSM strength.

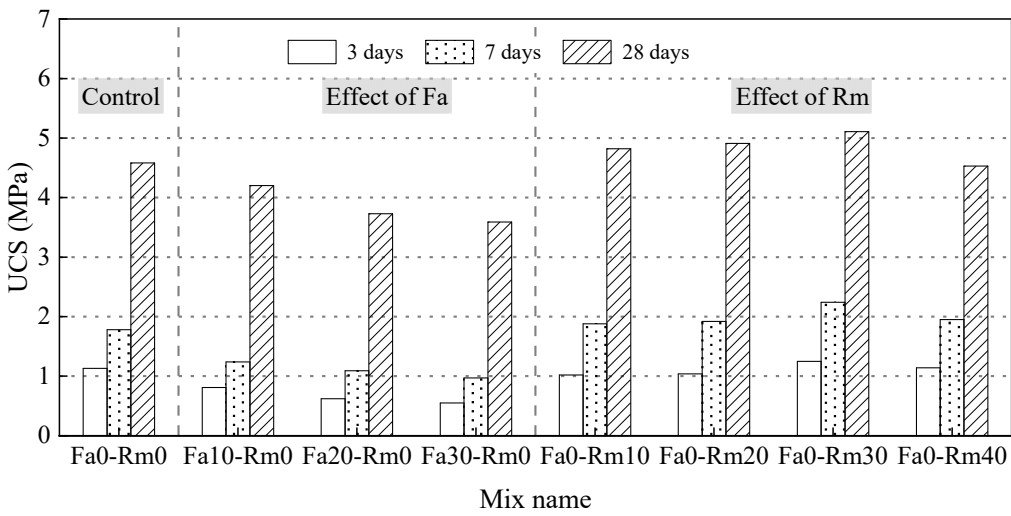

**Figure 5.** UCS of CLSM with independent replacement of Fa and Rm.

Figure 6 shows the strength change of CLSM for combined replacement with fly ash and red mud at 3, 7 and 28 days of curing age, respectively. Due to the presence of red mud, the CLSM strength tended to increase at first and then decrease with the growth of fly ash replacement. The strength of CLSM is very sensitive to the amount of fly ash replacement [30,31]. The test results showed that a 20% fly ash replacement amount was beneficial to the UCS development of CLSM. The deterioration effect with a high replacement amount of fly ash on strength was obvious in long-term curing. This showed that fly ash had a finite effect on the development of strength. Considering rapid construction, sufficient attention needs to be paid to the early strength (3 and 7 days) of CLSM [32]. With 20% fly ash replacement, the 30% red mud replacement was beneficial for the UCS development for 3 days and 7 days of the mixture. The small particle size and large specific surface area of red mud can reduce the hardening time and improve the early strength of the mixture by reducing the amount of free water in the slurry. At 28 days of curing, there was a significant strength difference at a high amount of fly ash replacement. Therefore, to meet the application requirements of high flowability and fast hardness of CLSM, Fa20-Rm30 was considered in this study as the optimal mix proportion for the combined replacement of fly ash and red mud.

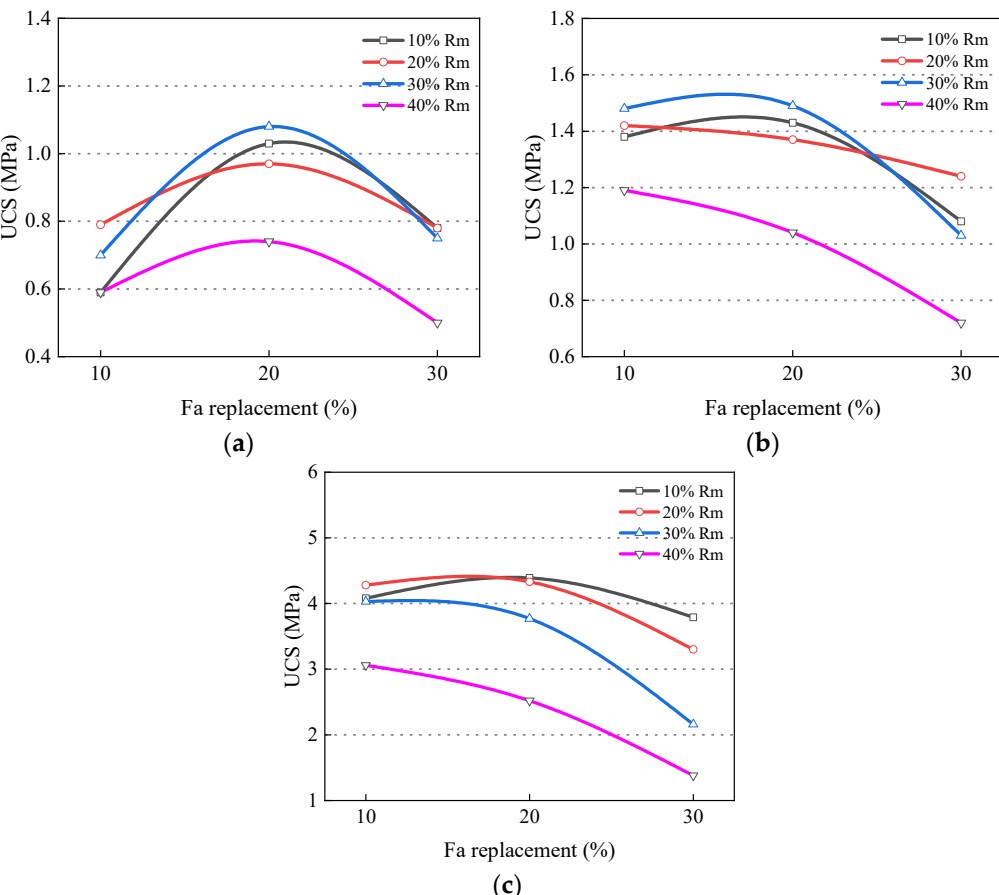

**Figure 6.** UCS of CLSM with co-replacement of Fa and Rm (**a**) 3 days; (**b**) 7 days; (**c**) 28 days.

### 3.3. XRD Analysis

The XRD patterns of Fa0-Rm0, Fa20-Rm0, Fa0-Rm30, Fa20-Rm30 are shown in Figure 7. Since the fine aggregate of CLSM was excavated soil, the quartz diffraction peak in the mixture was large. The hydration products were mainly C-S-H gels, and ettringite. There are two types of C-S-H gels: Jennite type and Tobermorite type. They are distinguished by the difference in Ca/Si ratios, which are larger in Jennite C-S-H gels [33]. The C-S-H diffraction peak in Figure 7 was the Jennite type, which had the lowest intensity in Fa20-Rm0 and the highest in Fa0-Rm30. The replacement of cement with fly ash reduced Ca/Si ratios in the mixture, which caused the C-S-H shift from Jennite type to Tobermorite type and consequently reduced the peak intensity.

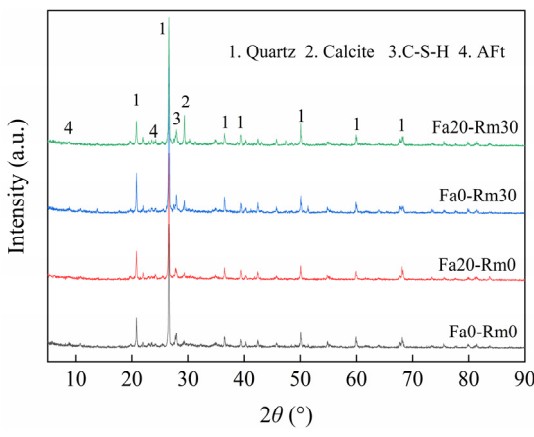

**Figure 7.** XRD patterns of CLSM at different mixes.

The addition of red mud to the mixture increased the alkalinity of the pore solution, while fly ash increased the concentration of $SO_4^{2-}$. This would promote the hydration reaction in the direction of AFt production and lead to an increase in its yield. Consequently, the intensity of the AFt diffraction peak was higher in the CLSM containing solid waste compared with Fa0-Rm0.

The source of calcite diffraction peak strength included two parts, one was soil, and the other was carbonization of hydration products. Compared with Fa0-Rm0 and Fa20-Rm0, Fa0-Rm30 and Fa20-Rm30 had a high calcite content. $Ca^{2+}$ combines with $CO_3^{2-}$ to produce a $CaCO_3$ crystal framework, which is slightly soluble in water. $CaCO_3$ precipitated out in the form of sediment and no longer participated in the reaction. Ban et al. [34] found that hydration products in polymers could coexist with $CaCO_3$. The development of strength depends on the strength of the hydration product itself and the compactness of the bonding particles. It should be noted that the amount of calcite in Fa20-Rm30 was significantly higher than in Fa0-Rm30. This indicated that more carbonate in the red mud was involved in the reaction after further reduction of the cement content [35].

### 3.4. SEM Analysis

SEM images can clearly show the microscopic morphology of CLSM. The effects of fly ash and red mud on the microstructure of CLSM were clarified. Fa0-Rm0, Fa20-Rm0, Fa0-Rm30 and Fa20-Rm30 were studied in SEM images, as shown in Figure 8. The structural differences in these microscopic images were in accordance with the changes in the strength of the components of the cementitious material.

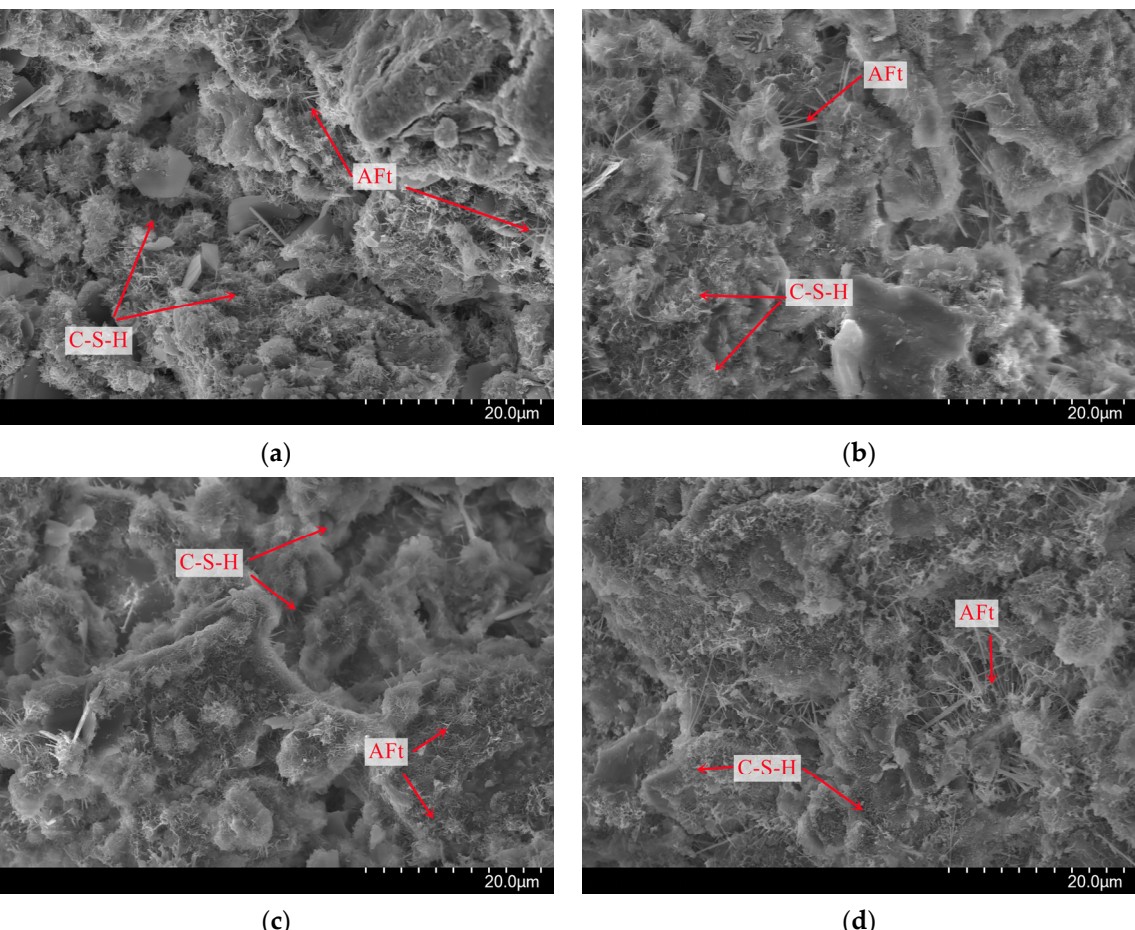

**Figure 8.** SEM images of CLSM at different mixes (**a**) Fa0-Rm0; (**b**) Fa20-Rm0; (**c**) Fa0-Rm30; (**d**) Fa20-Rm30.

The mixture was observed to be filled with a large amount of C-S-H gel and Aft, and formed a dense network structure in Fa0-Rm0. The strength of the mixture was improved by using C-S-H gels as the main cementing phase to connect other hydration products, which provided the bonding force for the soil particles [31]. Unlike the other groups, some hexagonal crystal structures were found in the mixture. This may have been caused by the partial conversion of AFt to AFm due to the decrease of SO2-4 concentration in the pore solution. A large number of C-S-H gels were found attached to the surface of the soil and fly ash particles in Fa20-Rm0. Fly ash provided good nucleation sites for C-S-H gels. The cement was no longer wrapped and the hydration process could proceed more deeply. The soil particles which can be seen in Fa0-Rm30 were covered by large areas of hydration products. This was due to the high alkalinity of the red mud promoting more C-S-H gels and AFt production. The hydration products can effectively reduce the harmful impact of macropores on the mixture structure. In addition, Ghalehnovi et al. [36] reported that the cementitious properties of red mud are not as good as cement, but its fine particles are more effective in filling the pores.

Cement in Fa20-Rm30 accounted for only 50% of all cementitious materials. Therefore, the total amount of hydration products was less than in the other groups. In addition to the gels, calcite also had a positive effect on structural compactness. The particle filling effect of fly ash and red mud can refine the pore structure. The high content of $SiO_2$ and $Al_2O_3$ in fly ash was excited by the high alkalinity of red mud. More hydration products were used to fill the internal voids of the CLSM to reduce the sample porosity, which was conducive to CLSM strength maintenance. In spite of the cement, the content was only half of that of Fa0-Rm0, and obvious pores did not appear (Figure 8d). Consequently, the synergistic effect of fly ash and red mud contributed to maintaining the good mechanical properties of CLSM mixtures even if the cement content was reduced.

## 4. Conclusions

In this study, fly ash and red mud as SCM to replace part of the cement, and excavated soil was used as fine aggregate to make CLSM with good self-leveling and self-compacting properties. The flowability, UCS, phase composition and microstructure of CLSM mixture were investigated by changing the replacement amount of fly ash and red mud. The main conclusions of this study are as follows:

(1) The effects of fly ash and red mud on CLSM performance were very different. Fly ash improved the flowability, while red mud was more beneficial for strength growth of the CLSM mixture. With the increase of fly ash replacement, the fluidity of CLSM first increased and then decreased, while the UCS of CLSM gradually decreased. When the fly ash replacement cement mass was 20%, the fluidity of CLSM reached an inflection point. With the increase of red mud replacement amount, the fluidity of CLSM gradually decreased until it was not easy to flow, and when the replacement amount was 30%, the UCS of CLSM showed an inflection point. When the cement was replaced by fly ash and red mud at the same time, the mixture mix proportion with optimal engineering performance was Fa20-Rm30. Its fluidity was 248 mm, and the UCS at 3, 7 and 28 days was 1.08, 1.49 and 3.77 MPa, respectively. The mixture met both high flowability and fast hardening requirements.

(2) Hydration products of CLSM cured for 28 days included C-S-H gels and ettringite. The C-S-H gels were the main cementing phase that connected other hydration products to form a dense mesh structure. The strong alkaline environment generated by red mud in contact with water not only promoted the hydration of cement but also had a positive effect on stimulating the activity of $SiO_2$ and $Al_2O_3$ in fly ash. The high content of carbonate in red mud promoted the carbonization reaction. The resulting calcium carbonate could coexist with other hydration products, increasing the strength of the mixture. Fly ash provided good nucleation sites for hydration products, and a large number of hydration products attached to the particle surface to refine the pore structure.

**Author Contributions:** Conceptualization, X.K. and Y.Z.; methodology, G.W. and S.R.; validation, G.W., Y.L. and M.L.; formal analysis, G.W.; investigation, S.R; resources, X.K and Y.Z.; data curation, S.R.; writing original draft, X.K. and G.W.; writing review, Y.Z.; supervision, X.K. All authors have read and agreed to the published version of the manuscript.

**Funding:** This research was funded by Key Research and Development Project of Shandong (Grant No. 2020CXGC011404).

**Institutional Review Board Statement:** Not applicable.

**Informed Consent Statement:** Not applicable.

**Data Availability Statement:** The data presented in this study are available on request from the corresponding author.

**Conflicts of Interest:** The authors declare no conflict of interest.

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
