# Peer review of "Utilization of Fly Ash and Red Mud in Soil-Based Controlled Low Strength Materials"

_coatings, doi:10.3390/coatings13050893_

Round 1

Reviewer 1 Report

Coatings

Manuscript ID: coatings-2381197

Utilization of Fly Ash and Red Mud in Soil-Based Controlled Low Strength Materials

by

Kong, Wang, Rong, Liang, Liu, Zhang

REFEREE’S COMMENTS

This is an interesting paper on a subject that should be of great interest to many readers. For location of comments, see the belows.

  1. Title:
  2. It should be more concise/compact.
  3. Abstract:
  4. There is no need to give introductory level sentences in this section.
  5. The abstract should be supported by quantitative findings.
  6. Introduction:
  7. The literature review could be strengthened. For example, see the papers Bulletin of Engineering Geology and the Environment 80, 8665-8676; Journal of materials in civil engineering 24 (7), 924-932; Applied clay science 127, 134-142; Journal of hazardous materials 321, 547-556; and many others already available in the literature.
  8. Last paragraph: The authors should clearly indicate the originality/novelty of their research. A separate paragraph would be better for this purpose.
  9. Materials and methods:
  10. How did the authors decide the mixing percentages?
  11. Results and Discussion:
  12. "3.1. Flowability analysis" refer relevant papers already available in the literature.
  13. Use gray colour-dotted-gridlines in the plot area of the Figures. Also, indicate the experimental points in the plots.
  14. "Results and Discussion" section was found to be well described. However, it would be required to discuss the findings in the present study with the results of other papers already available in the literature, with more details. Otherwise, it would be a kind of technical report rather than a scientific paper.
  15. Conclusions:
  16. The authors should strengthen the conclusions by referring the quantitative findings.
  17. In General:
  18. Quality of the Figures could be improved.
  19. Literature review should be extended.
  20. Check out the details of the references cited.

Best regards,

Fine

Reviewer 2 Report

Dear Editor,

The topic of the paper is interesting and suits the Journal of MDPI Coatings. However, a major revision is required before this manuscript is qualified to be published in this prestigious journal. The manuscript is needed to be revised grammatically. The authors are required to check the whole manuscript with a grammar specialist as it has several grammatical errors. Only after revising the manuscript based on the comments, the paper is suggested to be published in MDPI. Further information on various issues identified in the manuscript appears below:

1.    The introduction section needs to be revised. A paragraph should be dedicated to the importance of your work.

2.    The authors have done a great job on the literature review. However, the introduction needs more attention. More information on new materials related to the topic of this paper can be found here:

"Fracture properties evaluation of cellulose nanocrystals cement paste" Journal of Materials, 2020.

3.    Please provide a more detailed reasoning behind the behavior. The details should include the rigid numbers or percentages.

4.    Please indicate how many samples for each experiment have been used. Please revise the other experiments respectively.

5.    Please describe the process of each experiment. Also indicate the model of each tool that is used in the experiment. What is the accuracy of each machine? Please explain them accurately.

6.    Add error bars to your plots where possible.

7.    Conclusion needs more elaboration. Please use more sentences containing percentages and illustrate the main conclusions in the manuscript. Please paraphrase your results and discussions and use them in the conclusion part.

A grammatical check is needed.

Round 2

Reviewer 1 Report

-